# Antimicrobial and Antibiofilm Activities of Some Antioxidant 3,4-Dihydroxyphenyl-Thiazole-Coumarin Hybrid Compounds: In Silico and In Vitro Evaluation

**DOI:** 10.3390/antibiotics14090943

**Published:** 2025-09-18

**Authors:** Daniel Ungureanu, Gabriel Marc, Mihaela Niculina Duma, Radu Tamaian, Dan Cristian Vodnar, Brîndușa Tiperciuc, Cristina Moldovan, Ioana Ionuț, Anca Stana, Ovidiu Oniga

**Affiliations:** 1Department of Pharmaceutical Chemistry, “Iuliu Hațieganu” University of Medicine and Pharmacy, 41 Victor Babeș Street, 400012 Cluj-Napoca, Romania; daniel.ungureanu@elearn.umfcluj.ro (D.U.); cmoldovan@umfcluj.ro (C.M.); ionut.ioana@umfcluj.ro (I.I.); stana.anca@umfcluj.ro (A.S.); ooniga@umfcluj.ro (O.O.); 2Department of Organic Chemistry, “Iuliu Hațieganu” University of Medicine and Pharmacy, 41 Victor Babeș Street, 400012 Cluj-Napoca, Romania; marc.gabriel@umfcluj.ro; 3State Veterinary Laboratory for Animal Health and Safety, 1 Piața Mărăști Street, 400609 Cluj-Napoca, Romania; duma.mihaela-cj@ansvsa.ro; 4ICSI Business, National Institute for Research and Development for Cryogenic and Isotopic Technologies—ICSI Rm. Vâlcea, 4 Uzinei Street, 240050 Râmnicu Vâlcea, Romania; radu.tamaian@icsi.ro; 5Institute of Life Sciences, University of Agricultural Sciences and Veterinary Medicine, 3–5 Calea Mănăștur, 400372 Cluj-Napoca, Romania; dan.vodnar@usamvcluj.ro; 6Faculty of Food Science and Technology, University of Agricultural Sciences and Veterinary Medicine, 3–5 Calea Mănăștur, 400372 Cluj-Napoca, Romania

**Keywords:** thiazole, coumarin, catechol, hybrid compounds, antioxidant, antimicrobial, antibiofilm

## Abstract

**Background/Objectives**: In this study, we aimed to investigate the antimicrobial and antibiofilm activity of seven hydroxyphenyl-thiazolyl-coumarin hybrid compounds with antioxidant properties (**1a**–**g**), previously reported by our group. **Methods**: The compounds were evaluated in vitro through MIC, MBC, and MFC determinations, and percentage of biofilm (BF) inhibition and in silico, respectively, through molecular docking, molecular dynamics simulations, and ADMETox prediction. **Results**: All compounds showed antibacterial and antifungal activities. In terms of antibacterial activity, all the compounds were active on *Pseudomonas aeruginosa* (MICs = 15.62–31.25 μg/mL), *Enterococcus faecalis* (MICs = 15.62–31.25 μg/mL), and *Staphylococcus aureus* (MICs = 62.5–125 μg/mL). Regarding the antifungal activity, the effect against *Candida albicans* was similar to fluconazole (MIC = 15.62 μg/mL), compounds **1b** and **1g** being the most active against *Aspergillus brasiliensis* (MIC = 15.62 μg/mL). Furthermore, all compounds were both bactericidal and fungicidal. Regarding the antibiofilm activity, compounds **1d**–**g** showed superior *P. aeruginosa* BF inhibition compared to gentamicin. The in vitro results for the antibacterial activity were well correlated with the observations drawn in the molecular docking studies, where the best binding affinities (BAs) were observed against *P. aeruginosa* PAO1 GyrB subunit, and the molecular dynamics simulations confirmed the antibacterial mechanism of compounds **1a, 1b**, **1d**, **1f,** and **1g** through GyrB subunit inhibition. Regarding the antifungal activity, all compounds showed better BAs than fluconazole against CYP51 in all instances. ADMETox predictions concluded that all the compounds could have low gastrointestinal absorption and reduced risk of pharmacokinetic interactions. **Conclusions**: The investigated compounds bring novelty into the actual research due to their dual antibacterial and antibiofilm activity against biofilm-associated *P. aeruginosa* infections.

## 1. Introduction

Antimicrobials are very important molecules for modern society, given the increasing threat of antimicrobial resistance, especially due to the high-risk pathogens like *Pseudomonas aeruginosa*, *Acinetobacter baumannii*, and Enterobacteriales, resistant to carbapenems and to the latest cephalosporins. In this context, there is an important need for developing novel drugs based on several strategies available in medicinal chemistry, such as the design of new scaffolds, synthesis of hybrid compounds with synergistic activity, or repurposing of older molecules [1,2].

There is a lot of evidence that some compounds with antioxidant properties are able to modulate oxidative stress while exhibiting antimicrobial properties as well. This dual functionality is closely linked to the role of the reactive oxygen species (ROS), which are pivotal in microbial survival and pathogenicity. Recent findings have shown the importance of oxidative stress modulation in shaping the antimicrobial efficacy. Also, recent studies showed that exogenous nitric oxide (NO), produced by bacterial nitric oxide synthase (NOS), protects Gram-positive and Gram-negative bacteria (*P. aeruginosa*, *Staphylococcus aureus*, etc.) against oxidative stress and increases the bacterial resistance to a broad spectrum of antibiotics [3]. In the last years, a lot of studies have shown that fungal resistance is due to the capacity of fungal strains to form biofilms, considered critical in invasive fungal infections and associated with high mortality [4]. Some studies showed that only a few authorized antifungals are effective against the fungal biofilms, all of which have the capacity to induce ROS formation in fungal biofilm cells [4]. While low levels of ROS are essential for microbial signaling, excessive ROS accumulation disrupts redox homeostasis, leading to lipid peroxidation, protein denaturation, and DNA damage—ultimately resulting in microbial cell death [5]. Antioxidant compounds can either scavenge surplus ROS to weaken microbial defense systems or amplify ROS production to levels that overwhelm the microbial detoxification mechanisms (prooxidant effects) [6]. Targeting the oxidative pathways with antioxidant agents has shown promise in sensitizing resistant strains and enhancing the efficacy of the conventional antimicrobials [7]. In this context, new directions in the research of new antimicrobial drugs could be represented by the development of agents that act not only on classical microbial targets, but also through redox-based mechanisms.

Structurally, antimicrobials are very heterogeneous, containing various heterocycles, of which the five-membered ones are of high interest, which are found in several authorized antibacterials (e.g., ceftolozane, aztreonam, and dalfopristin) and antifungals (e.g., isavuconazole, abafungin, and voriconazole) [1,2].

The heterocyclic scaffolds such as thiazoles and triazoles are well-known for their antimicrobial and anti-inflammatory properties [8,9], while the benzopyrone (coumarin) scaffold attracts a lot of interest due to its multifaceted biological activities and particularly due to its pronounced antioxidant potential. The antioxidant capability by multiple mechanisms can mitigate cellular damage triggered by ROS [10,11,12] and can also play a critical role in the antimicrobial activity of the coumarin-based compounds.

Biofilms (BFs) are microbial communities constituted from bacterial or fungal colonies attached to an abiotic surface and covered by an extracellular matrix [1,13]. This ability enables the rise of complex infections involving both bacteria and fungi, which create significant challenges for treatment due to their heightened resistance to antimicrobial drugs [14].

One of the strategies to fight biofilm formation and avoid antimicrobial resistance is to facilitate the access of the antimicrobial drug to microorganisms inside the biofilm. Therefore, it is important to develop compounds with antibiofilm potential, even if they have low or no antimicrobial potential [1]. Currently, there is no approved antibiofilm drug, and the common practices include the removal of contaminated medical equipment, replacement of compromised implants, or the usage of long-term antibiotics if the replacement is not possible. There are several reported compounds with antibiofilm potential in the literature, including enzymes, quaternary ammonium salts, and plenty of natural compounds [15,16,17].

Thiazole-based compounds were reported in literature as antibiofilm agents, and their activity was attributed to the thiazole heterocycle, which was capable of BF inhibition [18,19]. Similarly, the coumarin heterocycle is an important structural element in anti-quorum-sensing (QS) and BF inhibitors [20]. The catechol moiety contributes to interfacial binding on adhesive proteins, thus providing an aid for better binding on BFs [21].

Starting from all these considerations, the current study aimed at investigating seven hydroxyphenyl-thiazolyl-coumarin compounds (**1a**–**g**, Figure 1) in terms of antibacterial activity against Gram-positive and Gram-negative bacteria, antifungal effects against Candida and Aspergillus strains, and antibiofilm activity against selected bacterial biofilms. The study represents an extension of our previous study, where we designed, synthesized, and evaluated the antioxidant activity of these compounds [22]. In this way, we aimed to study the possibility of synergism between the antioxidant and the antimicrobial and antibiofilm activities of these compounds. The hypothesis of this study was based on the idea of combining two different pharmacophores with antimicrobial properties into a single compound. The selected pharmacophores are found in authorized compounds with well-known mechanisms: the coumarin heterocycle from novobiocin, a compound with antibacterial activity due to the inhibition of the GyrB subunit; the azole heterocycle from reported DNA gyrase inhibitors with antibacterial activity and from the class of antifungal azoles with lanosterol 14α-demethylase inhibitory activity [23,24]. In order to establish their affinity towards the biological targets and a potential mechanism of action, the antimicrobial evaluation was completed with in silico molecular docking studies, while the antibacterial evaluation benefited from additional molecular dynamics studies.

Also, to complete the characterization of the compounds as potential drugs, we performed an in silico ADMETox study. This prediction is more economical and time-saving compared to the in vivo experimental determination of these parameters and could be useful for eliminating the molecules that are likely to fail in the early stages of drug discovery.

## 2. Results

### 2.1. Antimicrobial Activity

Compounds **1a**–**g** were tested for their antimicrobial activities, quantified as minimal inhibitory concentrations (MICs), minimal bactericidal concentrations (MBCs), and/or minimal fungicidal concentrations (MFCs), against Gram-negative bacterial strains, such as *Escherichia coli* (ATCC 25922), *Salmonella enteritidis* (ATCC 13076), *Salmonella typhimurium* (ATCC 14028), *Salmonella enteritidis* (isolated from a food source) and, *Pseudomonas aeruginosa* (ATCC 27853), and Gram-positive bacterial strains like *Listeria monocytogenes* (ATCC 13932), *Enterococcus faecalis* (ATCC 29212), and *Staphylococcus aureus* (ATCC 6538P), using ciprofloxacin as a reference. The antifungal activity was tested against *Candida albicans* (ATCC 10231) and *Aspergillus brasiliensis* (ATCC 16404), using fluconazole as a reference. The MIC, MBC, MFC, MBC/MIC ratio, and MFC/MIC ratio values are presented in Table 1, Table 2 and Table 3. All tested compounds showed bactericidal and fungicidal potential against all tested strains (MBC/MIC and MFC/MIC ratios < 4).

### 2.2. Antibiofilm Activity

Compounds **1a**–**g** were tested for their antibiofilm activity, quantified as percentage (%) of biofilm (BF) inhibition against *E. faecalis* ATCC 29212, *P. aeruginosa* ATCC 27853, *E. coli* ATCC 25922, and *S. typhimurium* ATCC 14028 biofilms, using gentamicin as reference. The percentages of BF inhibition are presented in Table 4.

### 2.3. Molecular Docking

The molecular docking study was performed on the bacterial DNA gyrase subunit B (GyrB) from *E. faecalis* V583, *S. aureus*, *P. aeruginosa* PAO1, and *E. coli* K-12, as well as on the fungal lanosterol 14α-demethylase cytochrome P450 (CYP51) from *C. albicans* and *Saccharomyces cerevisiae* YJM789. The selected bacterial and fungal targets differ between them in terms of the number of amino acid residues and selected sequences from the full-length proteins. Establishing the target affinity and binding mode, specific to each microorganism, may point out information regarding the efficacy and selectivity of the compounds on a specific bacterial and/or fungal strain.

The selection of GyrB as a molecular target for the antibacterial mechanism of compounds **1a**–**g** was motivated by the existing data regarding the potential of aminocoumarin compounds to inhibit the ATP-binding site of this subunit, considering that compounds **1a**–**g** share a similar structural profile [26,27].

The selection of CYP51 as a molecular target for the antifungal mechanism of compounds **1a**–**g** was motivated by the importance of this enzyme for the fungal cell membrane and because it is targeted by the antifungal azoles, considering that compounds **1a**–**g** also contain an azole (thiazole) in their structure [28].

#### 2.3.1. Target Selection

The extensive cross-check between The UniProt Knowledgebase (UniProtKB) and The Research Collaboratory for Structural Bioinformatics Protein Data Bank (RCSB PDB) websites resulted in the identification of the full-length sequences of the two molecular targets (bacterial GyrB and fungal CYP51) for most of the investigated microbial species (Table 5).

Among the identified molecular targets, only those with corresponding crystal structures in RCSB PDB were considered adequate for molecular docking (Table 6).

#### 2.3.2. Superposition and Morphing

Superposition was performed to detect possible structural deviations and whether they are attributable to differences in the quality of structures (resolution of RCSB PDB IDs combined with coverage of the sequence length) or mutations. Morphing provides a dynamic visualization of the conformational transitions of proteins and may provide an intuitive understanding of structural transitions that may impact function, especially if mutations are present in the binding pocket or the investigated structures are the *holo*-form of the protein.

The visual results of the GyrB and CYP51 crystal structures superposition and morphing analyses are available in Appendix B (Figure A1 and Figure A2) and Appendix A.

#### 2.3.3. Protein-Ligand Docking

We performed eight different P-LD runs for both GyrB and CYP51, meaning four runs for each target: 4GGL (*E. faecalis* V583), 6TCK (*S. aureus*), 7PTF (*P. aeruginosa* PAO1), and 7P2M (*E. coli* K12) for GyrB, respectively, 5V5Z (*C. albicans* SC5314), 5FSA (*C. albicans*), 5TZ1 (*C. albicans*), and 4WMZ (*S. cerevisiae* YJM789) for CYP51. The selected ligands were compounds **1a**–**g**, ciprofloxacin, and novobiocin for the bacterial targets, respectively, fluconazole for the fungal targets. The results of the eight P-LD runs were summarized in Table 7 and Table 8.

### 2.4. Molecular Dynamics Simulation

For all compounds and the co-crystalized ligand, the top binding conformation resulting from the molecular docking study was used to build the corresponding chimeric complexes with 7PTF, which were studied in molecular dynamics simulations.

The numeric data describing the evolution of the complexes of compounds with 7PTF were presented in Table 9. The data analyzed for the respective complexes included the RMSD of the heavy atoms of the ligand, the RMSD of the backbone, the radius of gyration of the atoms of the protein, and the hydrogen bonding between the ligand and the protein. The data was presented as average values. The evolution in time of the respective parameters during the simulations was presented as charts in Appendix A

### 2.5. ADMETox Prediction

When designing potential drug candidates, the virtual prediction of pharmacokinetic and toxicological properties (ADMETox) is a time-saving and facile process in the early drug discovery, preventing inadequate molecules from progressing further in development [37].

The pharmacokinetic descriptors considered were the GI absorption, BBB permeation, P-gp substrate, CYP1A2, CYP2C19, CYP2C9, CYP2D6, and CYP3A4 inhibition potential [38,39]. A BOILED-Egg graph for the prediction of GI absorption and BBB penetration capacity of compounds **1a**–**g** was generated.

The computed toxicologic descriptors were carcinogenicity, eye and skin irritation, hepatotoxicity, respiratory toxicity, reproductive toxicity, mitochondrial toxicity, nephrotoxicity, and acute oral toxicity [40,41,42,43,44,45,46,47]. Table 10 and Table 11 summarize the results of the ADMETox prediction carried out with SwissADME and admetSAR 2.0 web tools, respectively, with Toxtree 3.1.0 software.

The BOILED-Egg graph for the prediction of GI absorption and BBB penetration capacity of the compounds **1a**–**g** is illustrated in Figure 2.

## 3. Discussion

### 3.1. Molecular Docking

#### 3.1.1. Target Selection

According to Table 5, it can be observed that the bacterial target (GyrB) exhibited manually annotated full-length sequences in UniProtKB for *E. coli*, *P. aeruginosa*, *E. faecalis*, and *S. aureus*, and had corresponding crystal structures in RCSB PDB. In the case of *S. typhimurium*, only the reviewed full-length sequence was identified in UniProtKB, without any corresponding experimentally determined crystal structure. For *S. enteritidis* and *L. monocytogenes*, only unreviewed full-length sequences were identified in UniProtKB, but they lack experimentally determined structures, excluding their suitability for molecular docking.

For the fungal target (CYP51), for *C. albicans*, both the reviewed full-length sequences and corresponding crystal structures were identified (Table 5). In contrast, *A. brasiliensis* is represented in UniProtKB only by two unreviewed partial sequences and without any crystallographic data, thus excluding it from molecular docking. Therefore, to increase the fungal target pool, *S. cerevisiae* was considered a fitted candidate, a species with both a manually annotated full-length sequence in UniProtKB and a corresponding crystal structure in RCSB PDB.

According to Table 6, it can be observed that all four experimentally determined crystal structures of bacterial GyrB had high resolution (1.16 to 1.69 Å) and were free of mutations, all ensuring robustness of the molecular docking results. It should be noted that all four GyrB structures had less than half-sequence coverage, being cropped around the binding site of their co-crystallized ligands and covering the key functional regions for the current virtual screening (VS) setup. By contrast, all fungal CYP51 crystal structures had better to full-length sequence coverages, but with resolutions ranging from 2.00 to 2.90 Å. The lower resolution of the experimentally determined crystal (2.85 Å of 5FSA, and 2.90 Å of 5V5Z—both for *C. albicans* strains) might slightly reduce the accuracy of docking. Moreover, the crystal structures from two strains of *C. albicans* displayed mutations at specific residues (e.g., positions 6 and 221 in 5TZ1, and position 221 in 5FSA), which may induce local conformational changes that could affect ligand binding and should be carefully considered in the interpretation of P-LD results. Simultaneously, these mutations could be interesting case studies, with the reservation that the resolution of 5FSA (2.85 Å) is suboptimal for docking purposes, while the resolution of 5TZ1 (2.00 Å) is a more fitted choice. Within this context, choosing to use P-LD of a crystal structure from a species not covered by the wet-lab experiments (4WMZ from *S. cerevisiae*), but with a full-length sequence, good resolution (2.05 Å), and free of mutations, may increase the robustness of VS.

The presence of co-crystallized ligands (see Table 6) in all selected structures provides an essential benchmark for the accuracy of the current vs. setup, as the re-docking of co-crystallized ligands validates the docking protocol and scoring function. Notably, only one bacterial GyrB structure (7PTF from *P. aeruginosa* PAO1) was co-crystallized with a clinically relevant antibacterial drug (novobiocin), while all the fungal CYP51 structures were complexed with clinically relevant antifungals (including here, fluconazole, the antifungal used as a control compound in the wet-lab experiments). As can be observed in Table 6, ciprofloxacin, the antibacterial approved drug DB00537 [48] used as a control compound in the wet-lab experiments, was missing from the crystal structures selected for P-LD, as none of the crystal complexes deposited in RCSB PDB were fitted or relevant for the current study. Ciprofloxacin is a second-generation fluoroquinolone antibacterial approved drug [48], active against both Gram-negative and Gram-positive bacteria and an inhibitor of DNA gyrase (more active on the GyrA subunit than GyrB) and topoisomerase IV [48,49,50]. On the other hand, novobiocin is an aminocoumarin antibiotic, formerly approved as drug DB01051 [48] and currently “withdrawn from sale for reasons of safety or effectiveness” [51,52]. Novobiocin has a higher efficacy against the Gram-positive bacteria, while most of the Gram-negative bacteria are resistant [53]. Novobiocin binds to GyrB and blocks the ATPase activity [48,54,55], and it can also inhibit the activity of DNA topoisomerase I [56] and II [57]. The other two co-crystallized ligands of the selected experimentally determined crystal structures of bacterial GyrB were a pyrrolopyrimidine compound (CJC), a dual inhibitor of GyrB (4GGL, from *E. faecalis* V583) [35], and topoisomerase IV (4HYM, from *F. tularensis subsp. holarctica* LVS) [35], respectively, and a benzothiazole scaffold-based GyrB inhibitor (N1N), effective against both Gram-positive and Gram-negative bacteria, respectively, against *S. aureus* [30] and *E. coli* [31].

Careful selection of the best crystal structures of bacterial GyrB and fungal CYP51 complexed with their co-crystallized ligands ensures not only the robustness of P-LD results but also facilitates validation of computational predictions by ensuring consistency with experimentally observed binding modes.

#### 3.1.2. Protein-Ligand Docking

The four docking simulations against the bacterial GyrB—4GGL (*E. faecalis* V853), 6TCR (*S. aureus*), 7PTF (*P. aeruginosa* PAO1), and 7P2M (*E. coli* K12) (see Table 7)—indicated that all compounds **1a**–**g** exhibited superior binding affinities (BAs) to novobiocin in all instances and were comparable or superior to ciprofloxacin in almost all instances. Overall, the best P-LD scores were obtained for 7PTF (*P. aeruginosa*) and 6TCK (*S. aureus*), suggesting that these structures might provide the most reliable insights into ligand interactions, as all the GyrB experimentally determined crystal structures had a great resolution.

Compound **1e** showed excellent BAs on all four bacterial targets: 4GGL (BA = −9.0 kcal/mol—*E. faecalis*), 6TCK (BA = −9.3 kcal/mol—*S. aureus*), 7PTF (BA = −9.6 kcal/mol—*P. aeruginosa*), and 7P2M (BA = −9.1 kcal/mol—*E. coli*). Compound **1a** showed important BA on 6TCK (BA = −9.5 kcal/mol—*S. aureus*) and 7PTF (BA = 9.4 kcal/mol—*P. aeruginosa*).

The binding mode of compound **1e** at its peak affinity (BA = −9.6 kcal/mol against 7PTF—*P. aeruginosa*) versus the binding mode of ciprofloxacin (BA = −7.7 kcal/mol) and re-docking of the co-crystallized ligand (novobiocin, BA = −7.6 kcal/mol) is depicted hereinafter in Figure 3 and Figure 4. Both figures represent the graphic depiction of the best poses of the strongest binder (**1e**, colored using the Corey-Pauling-Koltun—CPK—standard coloring scheme in Figure 3A,B and Figure 4A–C, respectively, 2D image in Figure 4B_1_) against GyrB (7PTF—*P. aeruginosa* PAO1), by comparison with re-docking of co-crystallized ligand (novobiocin, colored in fuchsia in Figure 3A,B) versus the original co-crystallized ligand (novobiocin, colored in orange in Figure 3A,B) and ciprofloxacin (colored in lime in Figure 4C, respectively, 2D image in Figure 4C_1_), the control drug used in the wet-lab experiments. The target was depicted with the secondary structure drawn as cartoons (Figure 3A,B and Figure 4A–C) and electrostatic molecular surface (Figure 4A), while ligands are figured as sticks (Figure 3A,B and Figure 4A–C); for a clear image, the distant amino acid residues were hidden, and the view was cropped at a 10 Å distance from the co-crystallized ligand (novobiocin).

As observed in Figure 3 and Figure 4, the binding mode of compound **1e** was different from that of ciprofloxacin. The binding mode of **1e** to 7PTF (*P. aeruginosa* PAO1—Figure 4B,B_1_) indicates two types of interactions: (a) H-bonds (two with His118 and one with Gly119), and (b) steric interactions (strong steric interactions with Glu52, Asp75, and Val120, respectively, and a weak steric interaction with Gly121). The two strong H-bonds with His118 likely stabilize ligand positioning in the ATP-binding pocket of the target (see the position in Figure 4A while the H-bond with Gly119 suggests additional stabilization via backbone interactions, reinforcing the binding specificity. The three strong steric interactions with Glu52, Asp75, and Val120 create a hydrophobic or electrostatic fit, optimizing ligand binding, while the weak steric interaction with Gly121 suggests the existence of a flexible interaction that may contribute to the ligand adaptability in the binding pocket.

The binding mode of ciprofloxacin against 7PTF (*P. aeruginosa* PAO1—Figure 4C,C_1_) indicates two types of interactions: (a) one H-bond with Thr167, and (b) five weak steric interactions with Asn48, Ser49, Asp75, Arg78, and Ile80. The single H-bond of ciprofloxacin with GyrB is not as extensive or stabilizing as the multiple H-bonds stabilized between **1e** and GyrB. Steric interactions with polar (Asn, Ser, Asp, and Arg) and hydrophobic (Ile) residues suggest some degree of ligand accommodation in the binding site, but insufficient for a strong BA, leading to a reduced inhibitory activity of ciprofloxacin. Moreover, **1e** binds deeper inside the binding site of GyrB than ciprofloxacin (best viewed in Figure 4A), because it has a different binding mode compared to the other synthesized compounds, while novobiocin only blocks the entrance of the binding site due to its bulky spatial arrangement and its molecular size (Figure 3A,B).

Based on Table 7, it can be observed that there were minor variations between the four P-LD simulations against fungal CYP51, based on species, strain mutations, and resolution differences. Docking results for the low-resolution targets (5V5Z and 5FSA, both for *C. albicans*) and mutant strains (5FSA and 5TZ1, both for *C. albicans*) should be treated carefully and in a limited context. Notably, fluconazole, the antifungal drug used as a control in wet-lab experiments, scored as the weakest binder against the fungal CYP51 in all P-LD simulations. Conversely, **1e** scored as the strongest binder against the fungal CYP51 from *S. cerevisiae* (4WMZ—structure with high resolution and free of mutations), with a specific binding pattern (Figure 5 and Figure 6). Both figures represent the graphic depiction of the best poses of the strongest binder (**1e**, colored using the CPK standard coloring scheme in Figure 5A,B and Figure 6A,B, respectively, and the 2D image in Figure 6C) against CYP51 (4WMZ—*S. cerevisiae* YJM789), by comparison with re-docking of co-crystallized ligand (fluconazole, colored in fuchsia in Figure 5A,B) versus the original co-crystallized ligand (fluconazole, colored in orange in Figure 5A,B). The target was depicted with the secondary structure drawn as cartoons (Figure 5A,B and Figure 6A,B) and electrostatic molecular surface (Figure 6A), while ligands were figured as sticks (Figure 5A,B and Figure 6A,B); for a clear image, the distant amino acid residues were hidden and the view was cropped at a 10 Å distance from the co-crystallized ligand (fluconazole).

As observed in Figure 5 and Figure 6, the binding mode of compound **1e** was different from that of fluconazole. The binding mode of fluconazole against 4WMZ (*S. cerevisiae* YJM789) indicates three types of interactions: (a) a coordinating bond with heme iron, through the nitrogen atom of its triazole ring; (b) two H-bonds with HOH743 and H790 water molecules, leading to interactions with the hydrophilic amino acid residues in the proximity (Tyr126, Ser382, and Tyr140); and (c) steric interactions with hydrophobic amino acid residues (Met509, Thr318, Thr130, Phe134, Ile139, and Phe236) [34].

The binding mode of **1e** against 4WMZ (*S. cerevisiae* YJM789—Figure 6B,C) indicates two types of interactions: (a) two H-bonds with Pro462, and (b) steric interactions (three strong steric interactions with Val311, Thr318, and Ala476, respectively, and a weak steric interaction with Gly315). The two strong H-bonds with Pro462 have an overwhelming significance because proline is a rigid residue and forms a fixed kink in polypeptides. Therefore, compound **1e** is interacting with a rigid region in the binding site, increasing the binding stability. The strong steric interactions with Val311 and Thr318 contribute to a better hydrophobic or mixed interaction, while alanine (Ala476) is often involved in hydrophobic packing, increasing ligand stability in the binding site. The weak steric interaction with Gly315 suggests a certain degree of flexibility, allowing the ligand to adjust within the binding pocket.

### 3.2. Molecular Dynamics Simulation

The assays performed to investigate the antioxidant and antimicrobial activities of compounds **1a**–**g** highlighted a positive correlation between the antioxidant and antibacterial activities of these compounds. This observation represented the starting point for conducting a molecular dynamics simulation on the GyrB subunit from *P. aeruginosa* PAO1 (7PTF), for which the best BAs were obtained in the molecular docking study. The co-crystallized ligand, novobiocin, was kept for reference due to its ability to preferably bind to the GyrB subunit [48,54,55,56,57].

Compounds **1a**, **1d**, and **1f** exhibited a low movement in the binding pocket of 7PTF, with an average RMSD of the coordinates of their heavy atoms less than 0.21 nm, which was found for novobiocin, the reference inhibitor used in the present molecular dynamics study (Table 9). Intermediate stability was identified for compound **1g** (average RMSD = 0.27 nm) and a lower stability for compound **1b** (average RMSD = 0.37 nm), but with at least three moments of significant movement (Appendix A). Compounds **1c** (ethyl substituted) and **1e** (benzo[*f*] annulated) were found to be false positives resulting from the molecular docking study, with low stability in the targeted protein. This suggests that a different antibacterial mechanism may exist for compounds **1c** and **1e**, as discussed further.

A destabilization effect of the docked ligands **1b**, **1c**, and **1g** was identified, with an increased average backbone RMSD found when comparing the apo form of the protein (average backbone RMSD = 0.21 nm) to the complex with novobiocin (average backbone RMSD = 0.17 nm). This underlined the stabilization effect of novobiocin, but a similar effect could also be identified for compound **1a** (average backbone RMSD = 0.18 nm).

The analysis of the radius of gyration for the atoms of the protein 7PTF would not bring significant information for the evaluation of the stability of the studied complexes.

Regarding the hydrogen bonding pattern between the studied ligands and the host protein, the highest number of hydrogen bonds were found in the complexes of compounds **1a** (3.52 bonds/ns), **1d** (3.84 bonds/ns), **1f** (3.30 bonds/ns), and **1g** (3.90 bonds/ns), slightly different to novobiocin (4.31 bonds/ns). The lowest hydrogen bonding with the host protein was found for compounds **1b** (0.46 bonds/ns), **1c** (0.49 bonds/ns), and **1e** (1.96 bonds/ns). The low hydrogen-binding compounds to the protein were the same as those that exhibited the highest average ligand RMSD, which indicated that hydrogen bonding may play a significant role in the binding of the compounds in the targeted pocket of the protein.

For the well-bound compounds **1a**, **1d**, and **1f**, as indicated above, the inspection of the trajectories indicated a small accommodation movement in the targeted pocket in the first part of the simulation, with a further constant trajectory indicating the stability in time of the respective complexes. Compound **1g** showed an increasing movement in the last part of the simulation, thus suggesting a lower stability over time. By comparing compounds **1f** and **1g**, it could be considered that the position of the hydroxy group grafted on the coumarin heterocycle (6-OH in **1f** and 7-OH in **1g**) is significant for the stability of the complex.

### 3.3. Antimicrobial Activity

All compounds showed antibacterial and antifungal activity against the tested strains [58]. The activity also correlates with the results obtained in the molecular docking study, where the best BAs were observed against the GyrB subunits of *P. aeruginosa* (7PTF) and *S. aureus* (6TCK) (Table 7).

All compounds **1a**–**g** were active on *P. aeruginosa* (MICs = 15.62–31.25 μg/mL), *E. faecalis* (MICs = 15.62–31.25 μg/mL), and *S. aureus* (MICs = 62.5–125 μg/mL), the activity being similar or superior to ciprofloxacin (Table 1). The activity against *Salmonella* spp. (MICs = 31.25–125 μg/mL) and *L. monocytogenes* (MIC = 31.25 μg/mL) was inferior to ciprofloxacin (MIC = 15.62 μg/mL) (Table 1).

Moreover, the unsubstituted coumarin compound **1a** showed a similar activity to ciprofloxacin against *E. coli* (MIC = 15.62 μg/mL), while the activity of the 7-hydroxy substituted compound (**1g**) was two-fold higher against *P. aeruginosa* (MIC = 15.62 μg/mL) than the reference (MIC = 31.25 μg/mL).

The antibacterial activity decreased by introducing an O-alkyl rest in the eight position of the coumarin heterocycle (compounds **1b** and **1c**). The effect was more evident against the Gram-negative bacterial strains (*E. coli*, *Salmonella* spp., and *P. aeruginosa*—MICs = 62.5–125 μg/mL). The reduced antibacterial activity of compound **1b** may also be attributed to the unstable complex with the GyrB subunit, as observed in the molecular dynamics simulations.

All compounds showed antifungal activity against the tested strains [58]. The activity against *C. albicans* was similar to fluconazole (MIC = 15.62 μg/mL), with compounds **1b** (methyl-substituted) and **1g** (7-hydroxy-substituted) being two-fold more active (MIC = 7.81 μg/mL) than the reference. The same compounds were the most active against *A. brasiliensis* (MIC = 15.62 μg/mL).

Considering the lack of activity for fluconazole against *A. brasiliensis*, literature data for itraconazole were used for comparison reasons [25]. Itraconazole was selected because it was one of the co-crystallized ligands reported in the target selection phase of the molecular docking study, specifically in the CYP51 crystal structures from *C. albicans* and *S. cerevisiae* strains. The activity of the tested compounds was inferior to itraconazole (MIC = 4 μg/mL).

The antifungal activity of the tested compounds **1a**–**g** could be explained by their ability to form complexes between the coumarin carbonyl group and the azomethine nitrogen from the hydrazone linker.

According to the MBC/MIC and MFC/MIC ratios (Table 2 and Table 3), all compounds were considered bactericidal and fungicidal, since no ratio had a greater value than 4 [59]. The best bactericidal activity was found for compound **1g** against *E. coli* and **1a** against *S. typhimurium* (isolated from food sources). The best fungicidal activity was observed for compound **1a** against both strains and for compound **1e** against *C. albicans*.

Based on qualitative structure-activity relationships (SAR), the 7-hydroxy substitution (**1g**) induced overall the best antimicrobial activity, as seen in Figure 7. As previously stated, grafting ether groups in compounds **1b** and **1c** decreased the antibacterial activity, while the unsubstituted compound **1a** had a very modest activity against the *S. typhimurium* strain isolated from food sources but was similar to ciprofloxacin against *E. coli*.

The results of our previous study have shown that thiazolyl-coumarins **1a**–**g** present antioxidant activity (Table 12) [22]. The presence of the antioxidant effects in antibacterial compounds was demonstrated to be beneficial, particularly against *P. aeruginosa*, where the selection of pro-biofilm variants could be halted if oxidative stress is neutralized [60,61]. The obtained results in the antibacterial study were correlated with the antioxidant potential of compounds **1a**–**g**. Therefore, the compounds with the best antioxidant activity were the most active against *P. aeruginosa* and *S. aureus* [3], which confirms the observations stating that the capacity to neutralize ROS leads to an increased susceptibility of some bacterial strains (*P. aeruginosa* and *S. aureus*) to the antibacterial effects of these compounds. Several flavonoids, which are well-known natural antioxidants, have been demonstrated to properly inhibit the GyrB subunit. For example, kaempferol mediates the inhibition through its hydroxyl groups, while epigallocatechin gallate inhibits the GyrB subunit through its benzopyran ring that penetrates the active site [62].

Regarding the antifungal activity, we observed a negative correlation between the antifungal activity and the antioxidant capacity of the best antifungal compound **1b**, which had a lower capacity to reduce the oxidative stress [4]. This statement is supported by available data regarding the antifungal activity of miconazole against *C. albicans* biofilms, through the increase in endogenous ROS concentrations, and other experimental observations that correlate prooxidant and antifungal effects [4,63,64].

### 3.4. Antibiofilm Activity

Gentamicin was selected as a reference for this assay due to its ability to disrupt the production of bacterial virulence factors, including biofilms [65].

Each compound showed antibiofilm activity in different degrees against the tested strains, demonstrating the ability to penetrate these BFs. Regarding the activity against *E. faecalis* BF, there was a weak BF inhibition in all active cases, including gentamicin. The highest inhibition percentages were registered by compound **1f** (27.75–24.61%) at concentrations of 500–125 μg/mL, which were superior to gentamicin at the same concentrations (23.04–19.90%). Notable activity was also observed for compound **1e** at 250–125 μg/mL (26.18% BF inhibition).

The activity against *P. aeruginosa* BF was considered the most effective, since all compounds were active. At concentrations between 500 and 31.25 μg/mL, all compounds showed effective BF inhibitions over 50%. The differentiation between compounds was established at the concentration of 0.1 μg/mL, where all compounds still active (**1d**–**g**) showed superior inhibitory activity (64.62–23.93%) compared to gentamicin (3.77%). The most potent compounds were **1e** (64.62%), **1d** (54.45%), **1f** (46.02%), and **1g** (23.93%). The results are more significant in the context of gentamicin being an anti-Pseudomonas aminoglycoside [66].

All compounds showed effective inhibitory activity at 500–62.5 μg/mL against *E. coli* BF. While the antibiofilm activity of all compounds was comparable to gentamicin at 500–62.5 μg/mL, the reference kept its potency at all concentrations (87.03–79.25%), except for 0.1 μg/mL (15.71%).

Regarding the inhibition of *S. typhimurium* BF, neither the compounds (maximum inhibition percentage 46.23% for compound **1f** at 125 μg/mL) nor gentamicin (maximum inhibition percentage 45.09%) showed inhibition over 50%, thus considering them weak BF inhibitors. Compounds **1a**–**f** kept their activity until 0.1 μg/mL (22.21–5.05%), although inferior to gentamicin (27.93%). The weakest activity was registered by compound **1g**, which became inactive at concentrations lower than 62.5 μg/mL.

In both of the last two presented cases, there was a more potent BF inhibition at 125 μg/mL than at higher concentrations and also superior to gentamicin in one instance. Compound **1f** was superior to gentamicin at 125 μg/mL against *S. typhimurium* BF. The low antibiofilm activities against *E. faecalis* and *S. typhimurium* correlate with the low antibacterial activity registered against the same bacterial strains.

Overall, the best antibiofilm activity was observed against the *P. aeruginosa* BF, followed by *E. coli* BF, where although there were no effective inhibitors at concentrations under 62.5 μg/mL, all of them were effective over this threshold. This was followed by *S. typhimurium* BF, with many of the compounds active until the lowest concentration, although they were all weak BF inhibitors in all instances. Finally, the weakest activity was observed against the *E. faecalis* BF, where the compounds were active between 500 and 125 μg/mL, but only as weak inhibitors.

Based on the qualitative SAR studies in these compounds, the annulation of a supplementary aromatic ring, as seen in compound **1e**, along with the substitution in the sixth position of the coumarin heterocycle (compounds **1d** and **1f**), was the most favorable for the antibiofilm activity. On the other hand, substitution in the seventh position (compound **1g**) proved unfavorable, except for the activity against *P. aeruginosa* BF. The SAR studies are illustrated in Figure 8.

It is important to note that all compounds showed significant antibacterial and antibiofilm activities against *P. aeruginosa*. The in silico observations were greatly reproduced in the biological evaluations and additionally supported by the correlations with the antioxidant activity of these compounds. Further extrapolation of these results highlights compounds **1e** and **1g** as the best anti-Pseudomonas compounds. Compound **1e** showcased the best BF inhibition (64.62%) at the lowest concentration and equal antibacterial activity to ciprofloxacin (MIC = 31.25 μg/mL). Compound **1g** had superior activities compared to the references in both instances: 23.93% BF inhibition at the lowest concentration vs. 3.77% for gentamicin and MIC = 15.62 μg/mL vs. 31.25 μg/mL for ciprofloxacin.

Considering the clinical challenges of biofilm-associated *P. aeruginosa* infections, the increasing antimicrobial resistance of this pathogen, and the rarity of dual compounds with intrinsic antibacterial and antibiofilm activities against it, compounds **1a**–**g** provide a significant advantage in overcoming the current issues regarding *P. aeruginosa* infections [67,68].

### 3.5. ADMETox Prediction

Based on the computed pharmacokinetic descriptors, all compounds **1a**–**g** were predicted to have low GI absorption and no BBB penetration capacity, making them difficult to formulate for oral administration. This is in concordance with the previously predicted octanol-water partition coefficient implemented by Moriguchi (MLogP) and estimated solubility (ESOL) values. On the other hand, the combination of low GI absorption and the antimicrobial activity on enterobacterial and fungal strains motivates the possible selective indication of these compounds in gastrointestinal bacterial and fungal infections [69].

Another drawback is that these compounds are predicted as class III for acute oral toxicity (high toxicity). This may be related to the presence of the aliphatic secondary amine linked to the second position of the thiazole ring, which could be metabolized to a nitrosamine [47].

Additionally, all compounds were predicted to be non-P-gp substrates or CYP1A2, CYP2C19, CYP2D6, and CYP3A4 inhibitors. Compounds **1a**–**e** were predicted to be CYP2C9 inhibitors, while compounds **1f** and **1g** were predicted not to inhibit this isoenzyme. This may be because of the hydroxy groups grafted on their coumarin moieties, which may avoid any interaction with the isoenzyme CYP2C9, since it mostly catalyzes hydroxylation reactions [70]. The reduced potential of compounds **1a**–**g** to interfere with CYP450 isoenzymes represents an advantage in practice, when concomitant medications are used, compared to ciprofloxacin and fluconazole. Ciprofloxacin is a CYP3A4 and CYP1A2 inhibitor, while fluconazole is a strong CYP2C19 inhibitor and a moderate CYP2C9 and CYP3A4 inhibitor [71,72]. Virtually, compounds **1f** and **1g** are at no risk of producing drug-drug pharmacokinetic interactions related to metabolism.

## 4. Materials and Methods

### 4.1. Antimicrobial Activity

The compounds **1a**–**g** were tested for their antimicrobial activity, quantified as MIC, MBC, and MFC, using the microdilution method, according to a previously published protocol [73]. The initial concentration of the colonies was 10^8^ CFU/mL, corresponding to the McFarland 0.5 standard, which was further diluted to 10^5^ CFU/mL. The density of the inoculum was determined using a BioMérieux DensiCHECK (BioMérieux SA, Marcy-l’Étoile, France) densimeter.

Stock solutions of **1a**–**g** (1 mg/mL), ciprofloxacin, and fluconazole were prepared by dissolving the solid compounds in sterile DMSO (Merck KGaA, Darmstadt, Germany) and were stored at 4 °C. Double dilutions with final concentrations between 15.625 and 500 µg/mL were prepared in RPMI 1640 (Thermo Fischer Scientific Inc., Waltham, MA, USA) medium, starting from the stock solutions. The sample solutions (100 µg/mL) were added in the first rows of a 96-well plate, followed by 20 µL samples from the diluted solutions in the other wells.

The resazurin solution was used as an indicator dye in this study and was prepared according to a previously reported protocol [74]. To obtain a 0.015% solution, 15 mg resazurin powder (Merck KGaA, Darmstadt, Germany) was dissolved in 100 mL of sterile distilled water. The solution was sterilized by filtration and stored at 4 °C for a maximum of 2 weeks after preparation.

A volume of 10 μL of bacterial or fungal inoculum was added to each well, followed by 20 µL resazurin and incubation for 24 h at 37 °C or 48 h at 25 °C for *C. albicans* (ATCC 10231) and *A. brasiliensis* (ATCC 16404). MIC was defined as the lowest concentration that prevented a color shift. DMSO was used as a negative control because it does not have intrinsic antimicrobial properties, based on the given conditions in this study.

A 100 µL aliquot sample from the wells with no observed microbial growth was collected and further incubated for 24 h at 37 °C or for 48 h at 25 °C. MBC and MFC were defined as the lowest concentrations for which no growth was observed. All determinations were performed thrice.

### 4.2. Antibiofilm Activity

The strains were grown in a test tube containing 9 mL of sterile medium, NB (Nutrient Broth), MHB (Muller–Hinton Broth), and BHI (Brain Heart Infusion). The tubes with NB and BHI were incubated for 24 h at 37 °C for *E. coli* ATCC 25922, *E. faecalis* ATCC 29212, and *S. typhimurium* ATCC 14028. The tubes with MHB were incubated for 24 h at 37 °C for *P. aeruginosa* ATCC 27853. A loopful of inoculum was transferred to a growth agar medium. Plates were incubated with bacteria for 24 h at 37 °C. Optical microscopy confirmed bacterial morphology (Nikon ECLIPSE Ci-L, Tokyo, Japan).

Stock solutions of **1a**–**g** (1 mg/mL) and gentamicin, used as a positive control, were prepared by dissolving the solid compounds in sterile dimethylsulfoxide (DMSO) and were stored at 4 °C. Following the cultivation, 90 μL of medium was added to the wells, then 100 μL of sample was added to the first well, and serial 2-fold dilutions were made in the subsequent wells by transferring 100 µL from well to well, resulting in concentrations between 500 and 0.01 μg/mL.

The surplus of 100 μL from the last well of each row was discarded. Several colonies of each cultivated strain on the media were transferred into a sterile 9 mL culture medium. The turbidity was adjusted to match the turbidity of the McFarland 0.5 standard (10^8^ CFU/mL). Ten microliters of inoculum (10^8^ CFU/mL) suspension were added to each microplate well. The microplates were incubated for 24 h at 37 °C. The solvent used to prepare the sample was used as a negative control [75,76].

Finally, in order to quantify the biofilm, 150 μL of 30% acetic acid in water was added to each microplate well to solubilize the crystal violet. The microplate was incubated at room temperature for 20 min, then 150 μL of the solubilized crystal violet was transferred to a new flat-bottomed microplate. The optical density (OD) was measured in a plate reader at λ = 550 nm, and the percentage of biofilm inhibition was calculated using Equation (1) [75,77]:(1)$$Percentage\;\left(\%\right)inhibition=\frac{({OD}_{Negative\;control}-{OD}_{Sample})\times 100}{{OD}_{Negative\;control}}$$

BF inhibition was categorized from 0 to 100%. Values below 0% were recorded as 0% BF inhibition, those between 0% and 50% indicated weak anti-biofilm activity, and values exceeding 50% denoted effective biofilm inhibition. Any values surpassing 100% were reported as 100% BF inhibition [77]. Each determination was repeated three times.

### 4.3. Molecular Docking

#### 4.3.1. Target Selection

Prior to P-LD, an extensive cross-check between the UniProtKB website (https://www.uniprot.org/) [78,79] and the RCSB PDB website (http://www.rcsb.org/) [80,81] was performed to identify the best crystal structures for the two molecular targets (GyrB and CYP51) in correlation with the wet-lab experiments and taking into consideration the investigated microbial species. As the most promising starting point for a successful P-LD is the high-resolution (preferably < 2.0 Å) ligand-bound crystal structure of an enzyme (the *holo*-form) [82,83,84,85,86], this type of protein-ligand complex was prioritized in the applied search strategy.

#### 4.3.2. Superposition and Morphing

Superposition and morphing analyses of selected crystal structures (respectively, the selected RCSB PDB identifiers—RCSB PDB IDs) were performed with UCSF ChimeraX v.1.8 [87,88] to gain more insights into the structural differences across investigated variants of GyrB and CYP51.

#### 4.3.3. Protein-Ligand Docking

Separate P-LD runs were performed for each identified RCSB PDB ID using PyRx—Python Prescription v.0.9.2 [89] as a vs. interface and AutoDock Vina v.1.2.0 [36,90] as a docking engine. AutoDock Vina uses a genetic algorithm (GA) with local optimization for molecular docking, instead of a traditional GA [91]. Still, instead of Vina, it uses the Broyden-Fletcher-Goldfarb-Shanno (BFGS) optimization algorithm [92] to refine and cluster the best poses and a built-in empirical scoring function. All P-LD runs were performed in a search space delineated around the binding site of the co-crystallized ligand of each structure, and without exceeding a maximum volume of 27,000 Å^3^, as recommended by Vina developers [36]. To improve the docking engine’s accuracy, the exhaustiveness value was set to 800 (the default value of exhaustiveness is set to 8 in PyRx), disregarding the tremendous increase in the computation time.

### 4.4. Molecular Dynamics Simulation

The chimeric complexes were studied in molecular dynamics simulations using GROMACS 2024.5, under the CHARMM36 force field and the TIP4P water model within an orthorhombic simulation box [93,94,95]. Ligand parameters were generated using the CGenFF. The preparation of the systems included solvation, neutralization with counterions, and energy minimization according to previously established procedures [96,97,98,99].

Each system was simulated for 100 nanoseconds (ns) to observe the dynamic structural evolution of the complexes. Computations were performed under Debian Linux 12 on a setup based on an AMD Ryzen™ 9 7900 CPU and a NVIDIA RTX 3060 GPU, with CUDA 12.8. Visualizations of the evolution of the simulated systems were made with VMD 1.9.4 [100].

### 4.5. ADMETox Prediction

The in silico prediction of the pharmacokinetic and toxicologic profiles (ADMETox) for compounds **1a**–**g** was performed using SwissADME, admetSAR 2.0 web tools, and Toxtree 3.1.0 open-source software [39,46,47,101].

Except for the acute oral toxicity, all toxicological descriptors were computed using the admetSAR 2.0 web tool. The acute oral toxicity was predicted using Toxtree 3.1.0 software, using the revised Cramer decision tree method [40,41,42,43,44,45,46,47].

## 5. Conclusions

In this study, seven hydroxyphenyl-thiazolyl-coumarin antioxidants (**1a**–**g**), previously reported by our research group, were evaluated for their antimicrobial and antibiofilm activities against selected bacterial and fungal strains, based on the idea of clubbing two different pharmacophores with antimicrobial properties into a single compound. The antimicrobial evaluation was completed with an in silico molecular docking study, in order to determine their affinity towards the selected biological targets, the bacterial GyrB and the fungal CYP51. The antibacterial activity was further explored through molecular dynamics simulations. Lastly, the characterization of the compounds as potential drugs was completed with an in silico ADMETox prediction assay.

According to the results from the antimicrobial assay, all compounds showed antibacterial and antifungal activity against the tested strains. In terms of antibacterial activity, all compounds showed important activity against *P. aeruginosa*, *E. faecalis*, and *S. aureus*, with compound **1g** being two-fold more active against *P. aeruginosa* than ciprofloxacin. Compound **1a** showed comparable activity to the reference against *E. coli*.

Regarding the antifungal activity, all compounds showed similar activity to fluconazole against *C. albicans*, except for compounds **1b** and **1g**, which were twice as active. The same compounds were the most active against *A. brasiliensis*. Nevertheless, all compounds were determined to be both bactericidal and fungicidal.

Based on the correlations drawn in this study between the antioxidant and antimicrobial activities of compounds **1a**–**g**, the best antioxidant activity was present in those compounds that were the most active against *P. aeruginosa* and *S. aureus*, while the antioxidant activity was negatively correlated with the antifungal activity, as seen in compound **1b**.

All compounds registered antibiofilm activity in various degrees, demonstrating biofilm penetration capacities. The most important activity was registered against the *P. aeruginosa* BF, where four compounds (**1d**–**g**) at 0.1 μg/mL showed more potent BF inhibition than gentamicin at the same concentration. Notable results were also registered for the inhibition of *E. coli* BF, where all compounds showed effective BF inhibition at 500–62.5 μg/mL. Regarding the activity against the *E. faecalis* and *S. typhimurium* BFs, the compounds and the reference showed weak BF inhibition, although six out of seven compounds (**1a**–**f**) were still active against *S. typhimurium* BF at the lowest concentration. The lowest percentages were observed against the *E. faecalis* BF. The combined potent antibiofilm and antibacterial activities against *P. aeruginosa* of compounds **1a**–**g** (particularly compounds **1d**–**g**) represent a promising premise for further exploration in the therapy of these compounds.

Favorable substitutions for the antibiofilm activity included the annulation of a supplementary aromatic ring on the coumarin heterocycle (compound **1e**) and substitution in the sixth position of the same heterocycle (bromine for compound **1d** and hydroxy for compound **1f**).

The in vitro antimicrobial activity was well correlated with the observations drawn in the molecular docking study. All compounds showed superior BAs to novobiocin against all four GyrB subunits and comparable BAs to ciprofloxacin. The best BAs were expressed against 7PTF (*P. aeruginosa* PAO1) for all compounds. Similarly, all compounds showed superior BAs to fluconazole in all instances. The molecular dynamics simulations conducted on the chimeric complexes between compounds **1a**–**g** and 7PTF concluded that these compounds, except for **1c** and **1e**, act as antibacterials by targeting the GyrB subunit, given the stability of these complexes. Compounds **1c** and **1e** may have a different antibacterial mechanism.

Based on the ADMETox predictions, all compounds were predicted to have low GI absorption and no BBB penetration capacity. Low GI absorption may provide an advantage in proposing compounds **1a**–**g** as antimicrobials against GI bacterial and fungal infections, due to the reduced risk of possible adverse effects. Additionally, compounds **1a**–**g** were predicted to have a lower risk of pharmacokinetic interactions compared to ciprofloxacin and fluconazole.

Finally, the investigated compounds bring novelty into the actual research due to their dual antibacterial and antibiofilm activity against biofilm-associated *P. aeruginosa* infections, providing an advantage in the current context of increasing antimicrobial resistance and hardships in therapy.

## Figures and Tables

**Figure 1 antibiotics-14-00943-f001:**
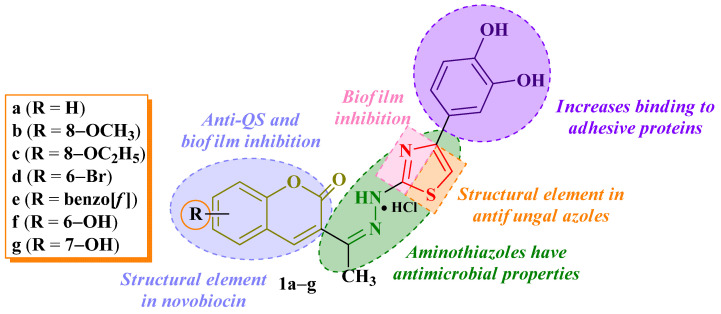
Chemical structures of compounds **1a**–**g** and research hypothesis as antimicrobials and antibiofilm agents [22].

**Figure 2 antibiotics-14-00943-f002:**
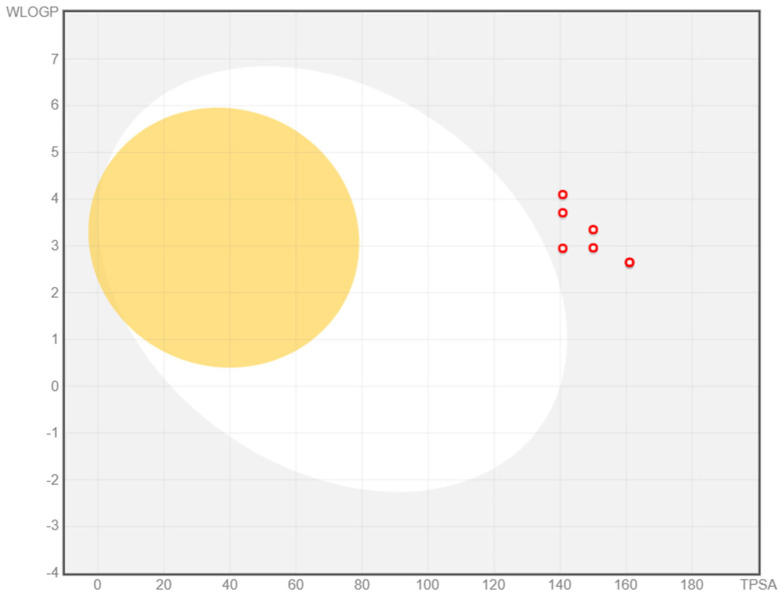
BOILED-Egg graph for the prediction of GI absorption and BBB penetration capacity of the compounds **1a**–**g** (as red circles).

**Figure 3 antibiotics-14-00943-f003:**
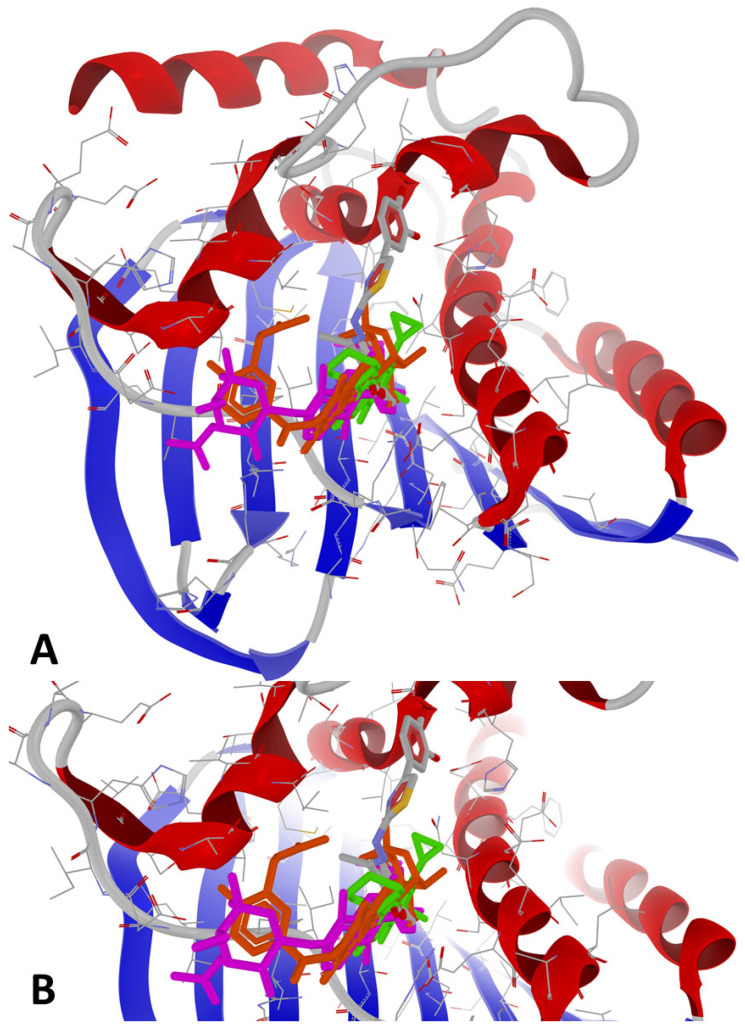
General view (**A**) and zoomed view (**B**) of the best poses of the strongest binder (**1e**, colored using the CPK standard coloring scheme) against 7PTF (*P. aeruginosa* PAO1), by comparison with re-docking of co-crystallized ligand (novobiocin, colored in fuchsia) versus the original co-crystallized ligand (novobiocin, colored in orange) and ciprofloxacin (colored in lime).

**Figure 4 antibiotics-14-00943-f004:**
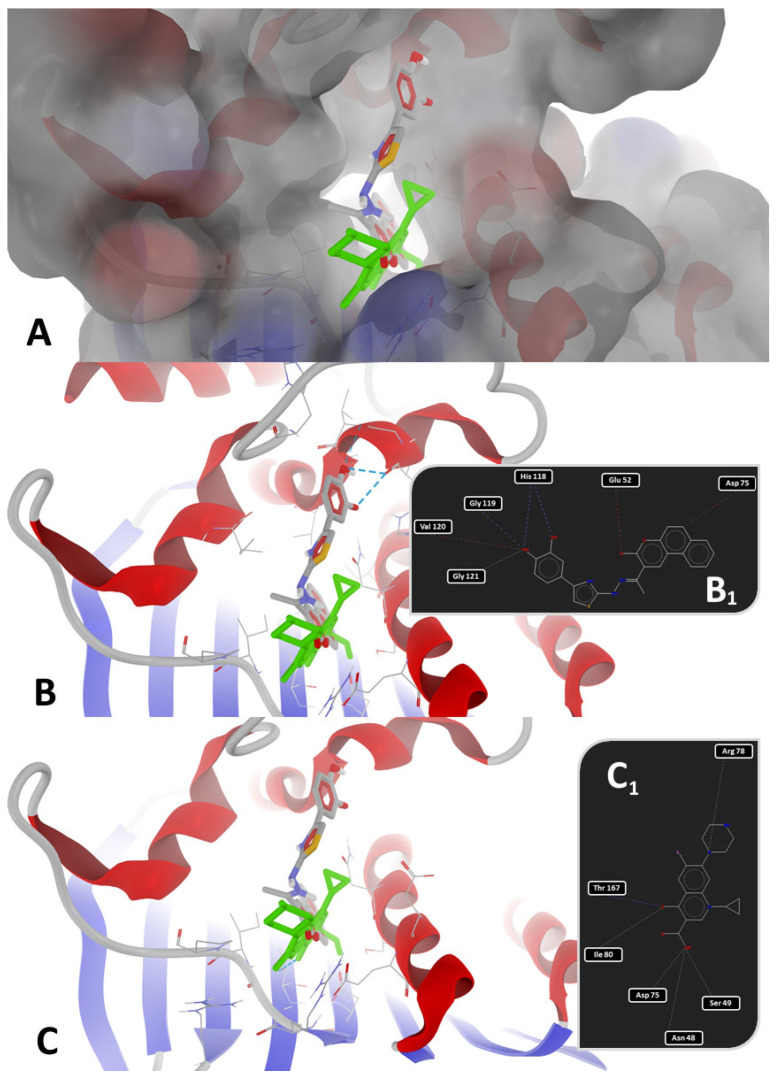
Detailed views of binding modes of **1e** versus ciprofloxacin in the binding pocket of 7PTF (*P. aeruginosa* PAO1): (**A**)—depiction of target with electrostatic molecular surface showing the ligands buried in the binding pocket; (**B**)—full 3D depiction of both poses with the three H-bonds of **1e** drawn as dashed blue lines; (**B_1_**)—simplified 2D depiction of **1e** with H-bonds drawn as dashed blue lines, strong steric interactions drawn as dashed red lines, and a weak steric interaction drawn as a dashed grey line); (**C**)—full 3D depiction of both poses with the single H-bond of ciprofloxacin drawn as a dashed blue line; (**C_1_**)—simplified 2D depiction of ciprofloxacin with the H-bond drawn as a dashed blue line, and weak steric interactions drawn as dashed grey lines.

**Figure 5 antibiotics-14-00943-f005:**
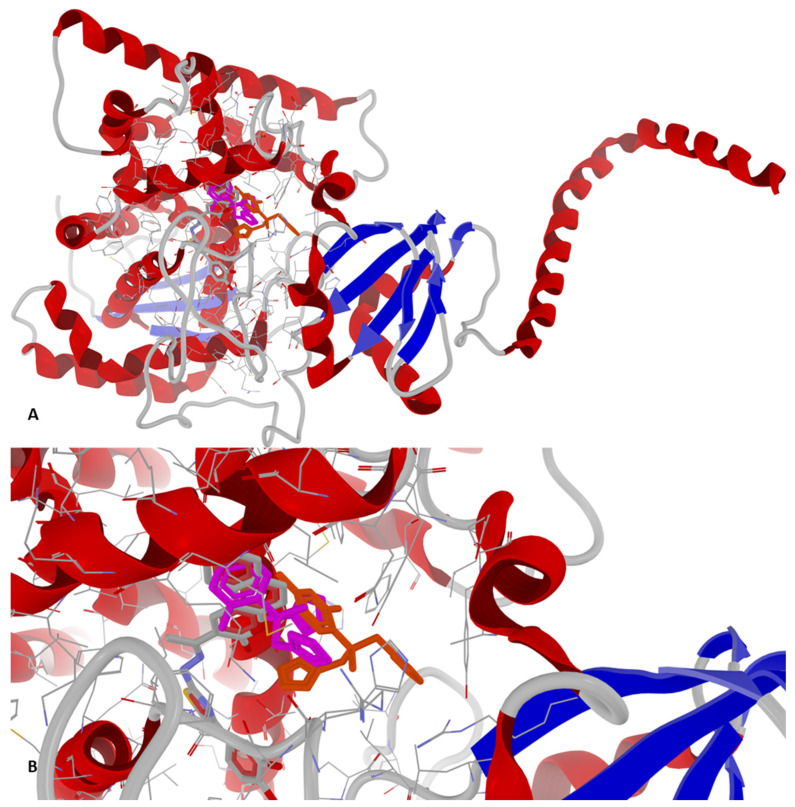
General view (**A**) and zoomed view (**B**) of the best poses of the strongest binder (**1e**, colored in CPK colors) against 4WMZ (*S. cerevisiae* YJM789), by comparison with re-docking of co-crystallized ligand (fluconazole, colored in fuchsia) versus the original co-crystallized ligand (fluconazole, colored in orange).

**Figure 6 antibiotics-14-00943-f006:**
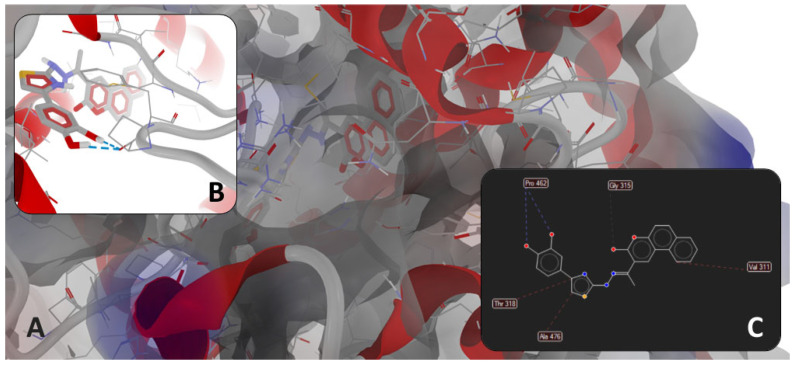
(**A**): detailed view of the binding pocket with docked **1e** deeply buried inside the target enzyme (molecular surface of target has 40% transparency and is colored according to the parental color of the residues). Detailed views of binding modes of **1e** in the binding pocket of 4WMZ (*S. cerevisiae* YJM789): (**B**)—full 3D depiction with H-bonds drawn as dashed blue lines; (**C**)—simplified 2D depiction with H-bonds drawn as dashed blue lines, strong steric interactions drawn as dashed red lines, and a weak steric interaction drawn as a dashed grey line.

**Figure 7 antibiotics-14-00943-f007:**
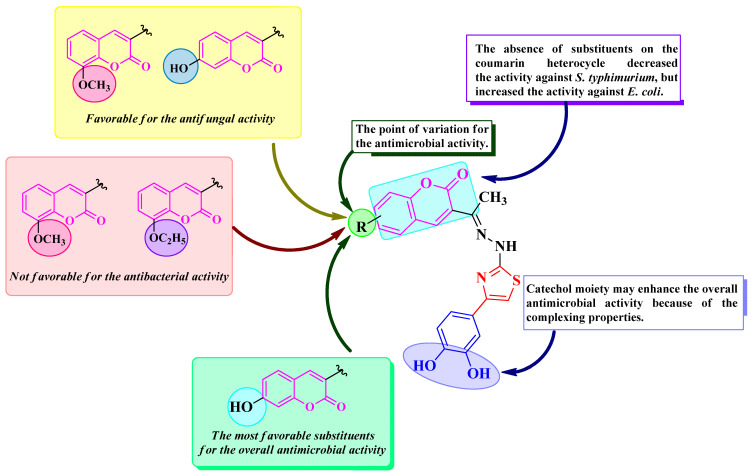
Qualitative SAR study of the antimicrobial activity of the novel catechol–thiazolyl–hydrazonoethyl–coumarin hybrid compounds **1a**–**g**.

**Figure 8 antibiotics-14-00943-f008:**
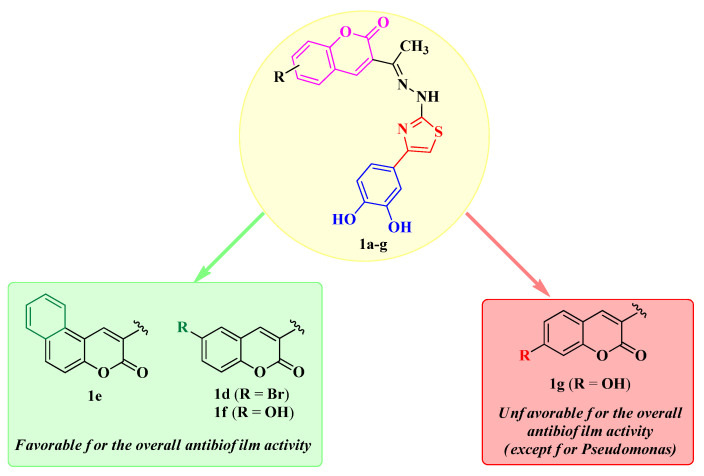
Qualitative SAR study of the antibiofilm activity of the novel catechol–thiazolyl–hydrazonoethyl–coumarin hybrid compounds **1a**–**g**.

**Table 1 antibiotics-14-00943-t001:** The MIC (μg/mL) and MBC (μg/mL) values of compounds **1a**–**g** against the tested bacterial strains.

Comp.	*E. coli* (ATCC 25922)		*S. enteritidis* (ATCC 13076)		*S. typhimurium* (ATCC 14028)		*S. typhimurium* (Food Isolate)		*P. aeruginosa* (ATCC 27853)		*L. monocytogenes* (ATCC 13932)		*E. faecalis* (ATCC 29212)		*S. aureus* (ATCC 6538P)	
	MIC	MBC	MIC	MBC	MIC	MBC	MIC	MBC	MIC	MBC	MIC	MBC	MIC	MBC	MIC	MBC
**1a**	**15.62**	**31.25**	62.5	62.5	62.5	62.5	125	62.5	**31.25**	**31.25**	31.25	**31.25**	**62.5**	**125**	**15.62**	**15.62**
**1b**	125	125	125	125	125	250	62.5	125	**31.25**	**62.5**	31.25	62.5	125	250	**15.62**	**31.25**
**1c**	125	125	125	250	125	125	62.5	125	**31.25**	**62.5**	31.25	**31.25**	125	**125**	**15.62**	**31.25**
**1d**	62.5	125	62.5	125	62.5	125	31.25	62.5	**31.25**	**62.5**	31.25	62.5	**62.5**	**62.5**	**15.62**	**31.25**
**1e**	62.5	62.5	62.5	125	62.5	125	31.25	62.5	**31.25**	**62.5**	31.25	62.5	**62.5**	**125**	**15.62**	**31.25**
**1f**	62.5	125	62.5	125	62.5	125	31.25	62.5	**31.25**	**31.25**	31.25	62.5	**62.5**	**125**	**15.62**	**31.25**
**1g**	125	62.5	125	250	125	250	31.25	62.5	**15.62**	**31.25**	31.25	**31.25**	125	250	31.25	62.5
**D**	Bacterial growth in all wells															
**M+**		+		+		+		+		+		+		+		+
**CPF**	15.62	31.25	15.62	31.25	15.62	31.25	15.62	31.25	31.25	62.5	15.62	31.25	125	250	15.62	31.25

Legend: Comp—compound; D—DMSO; M+—inoculum control; CPF—ciprofloxacin.

**Table 2 antibiotics-14-00943-t002:** The MBC/MIC ratios for compounds **1a**–**g** against the tested bacterial strains.

Compound	MBC/MIC Ratio							
	*E. coli* (ATCC 25922)	*S. enteritidis* (ATCC 13076)	*S. typhimurium* (ATCC 14028)	*S. typhimurium* (Food Isolate)	*P. aeruginosa* (ATCC 27853)	*L. monocytogenes * (ATCC 13932)	*E. faecalis* (ATCC 29212)	*S. aureus* (ATCC 6538P)
**1a**	2	1	1	0.5	1	1	2	1
**1b**	1	1	2	2	2	2	2	2
**1c**	1	2	1	2	2	1	1	2
**1d**	2	2	2	2	2	2	1	2
**1e**	1	2	2	2	2	2	2	2
**1f**	2	2	2	2	1	2	2	2
**1g**	0.5	2	2	2	2	1	2	2
**CPF**	2	2	2	2	2	2	2	2

Legend: CPF—ciprofloxacin; MBC/MIC ratios < 4—bactericidal; MBC/MIC ratios ≥ 4—bacteriostatic.

**Table 3 antibiotics-14-00943-t003:** The MIC (μg/mL) and MFC (μg/mL) values and MFC/MIC ratios of compounds **1a**–**g** against the tested fungal strains.

Compound	*C. albicans* (ATCC 10231)			*A. brasiliensis* (ATCC 16404)		
	MIC	MFC	MFC/MIC	MIC	MFC	MFC/MIC
**1a**	15.62	15.62	1	31.25	31.25	1
**1b**	7.81	15.62	2	15.62	31.25	2
**1c**	15.62	31.25	2	31.25	62.5	2
**1d**	15.62	31.25	2	31.25	62.5	2
**1e**	15.62	15.62	1	31.25	62.5	2
**1f**	15.62	31.25	2	31.25	62.5	2
**1g**	7.81	15.62	2	15.62	31.25	2
**DMSO**	Fungal growth in all wells					
**M+**		+			+	
**Fluconazole**	15.62	31.25	2	>250	>250	ND
**Itraconazole** [25]	-	-	-	4	-	-

Legend: M+—inoculum control; DMSO—dimethylsulfoxide; MFC/MIC ratios < 4—fungicidal; MBC/MIC ratios ≥ 4—fungistatic; “-”—not tested.

**Table 4 antibiotics-14-00943-t004:** Percentage (%) of BF inhibition against *E. faecalis* ATCC 29212, *P. aeruginosa* ATCC 27583, *E. coli* ATCC 25922, and *S. typhimurium* ATCC 14028 biofilms.

Concentration (μg/mL)	C_1_ = 500	C_2_ = 250	C_3_ = 125	C_4_ = 62.50	C_5_ = 31.25	C_6_ = 15.62	C_7_ = 7.81	C_8_ = 2.60	C_9_ = 1.30	C_10_ = 0.60	C_11_ = 0.20	C_12_ = 0.10
**Compound**	**BF inhibition (%)—*E. faecalis* ATCC 29212**											
**1a**	13.61	–	–	–	–	–	–	–	–	–	–	–
**1b**	15.18	19.90	21.47	–	–	–	–	–	–	–	–	–
**1c**	10.47	18.32	19.90	12.04	–	–	–	–	–	–	–	–
**1d**	4.19	10.47	16.75	–	–	–	–	–	–	–	–	–
**1e**	13.61	**26.18**	**26.18**	–	–	–	–	–	–	–	–	–
**1f**	**27.75**	**24.61**	**24.61**	–	–	–	–	–	–	–	–	–
**1g**	–	12.04	18.32	–	–	–	–	–	–	–	–	–
**Gentamicin**	23.04	21.47	19.90	15.18	–	–	–	–	–	–	–	–
	**BF inhibition (%)—*P. aeruginosa* ATCC 27583**											
**1a**	96.24	96.24	96.31	94.95	71.81	–	–	–	–	–	–	–
**1b**	95.27	96.63	96.57	94.56	75.18	56.07	55.16	–	–	–	–	–
**1c**	96.70	96.76	96.57	83.22	47.97	43.50	43.76	49.59	–	–	–	–
**1d**	95.14	96.37	96.37	77.51	58.40	41.62	25.48	49.07	45.83	71.81	**67.86**	**54.45**
**1e**	96.37	96.70	96.50	81.40	51.66	11.74	49.98	28.20	54.32	53.73	**64.49**	**64.62**
**1f**	96.70	96.70	95.66	93.00	69.35	40.26	6.04	46.09	31.44	55.61	**53.28**	**46.02**
**1g**	96.18	96.44	95.20	95.85	84.38	70.97	81.92	14.14	80.63	26.32	**76.54**	**23.93**
**Gentamicin**	96.50	96.70	96.70	96.57	88.73	94.95	95.79	95.20	94.95	96.11	50.82	3.77
	**BF inhibition (%)—*E. coli* ATCC 25922**											
**1a**	81.06	75.62	83.92	72.76	–	–	–	–	–	–	–	–
**1b**	82.36	83.92	86.25	73.28	–	–	–	–	–	–	–	–
**1c**	84.17	85.73	85.99	83.40	–	–	–	–	–	–	–	–
**1d**	81.06	84.17	86.77	77.43	–	–	–	–	–	–	–	–
**1e**	84.69	86.25	86.77	78.73	–	–	–	–	–	–	–	–
**1f**	85.47	86.51	86.25	83.66	–	–	–	–	–	–	–	–
**1g**	84.69	84.43	85.73	82.88	–	–	–	–	–	–	–	–
**Gentamicin**	85.99	87.03	86.77	87.03	82.88	85.21	83.40	83.40	84.43	79.25	80.28	15.71
	**BF inhibition (%)—*S. typhimurium* ATCC 14028**											
**1a**	25.64	15.35	38.23	27.93	5.05	7.34	13.06	18.78	16.49	14.20	13.06	7.34
**1b**	37.08	31.36	35.94	29.08	13.06	15.35	22.21	22.21	25.64	26.79	18.78	18.78
**1c**	34.80	35.94	41.66	32.51	13.06	15.35	23.36	23.36	24.50	32.51	22.21	22.21
**1d**	26.79	24.50	35.94	32.51	9.63	3.91	24.50	27.93	26.79	32.51	24.50	21.07
**1e**	40.51	**39.37**	42.80	39.37	3.91	3.91	10.77	22.21	26.79	16.49	25.64	5.05
**1f**	**45.09**	**40.51**	**46.23**	39.37	26.79	23.36	32.51	25.64	30.22	27.93	29.08	10.77
**1g**	30.22	30.22	17.64	29.08	–	–	–	–	–	–	–	–
**Gentamicin**	43.95	38.23	45.09	40.51	43.95	35.94	41.66	38.23	39.37	39.37	29.08	27.93

**Table 5 antibiotics-14-00943-t005:** Molecular targets and availability of their sequences (in UniProtKB) and existence of the experimentally determined crystal structures (in RCSB PDB).

Bacterial Target: DNA Gyrase Subunit B (GyrB)			
UniProtKB	Annotation	Organism	RCSB PDB
**Q839Z1**	reviewed	*E. faecalis* (strain ATCC 700802/V583)	Yes
**P0A0K8**	reviewed	*S. aureus*	Yes
**Q9I7C2**	reviewed	*P. aeruginosa* (strain ATCC 15692/DSM 22644/CIP 104116/JCM 14847/LMG 12228/1C/PRS 101/PAO1)	Yes
**P0AES6**	reviewed	*E. coli* (strain K12; ATCC 23724)	Yes
**A0A4U8JAX8**	unreviewed	*S. enteritidis*	No
**P0A2I3**	reviewed	*S. typhimurium* (strain LT2/SGSC1412/ATCC 700720)	No
**Q8YAV7**	unreviewed	*L. monocytogenes* serovar 1/2a (strain ATCC BAA-679/EGD-e)	No
**Fungal target: lanosterol 14α-demethylase cytochrome P450 (CYP51)**			
**UniProtKB**	**Annotation**	**Organism**	**RCSB PDB**
**P10613**	reviewed	*C. albicans* (strain SC5314/ATCC MYA-2876)	Yes
**A6ZSR0**	reviewed	*S. cerevisiae* YJM789	Yes
**A0A1L9U4P7 (AA: 524)**	unreviewed	*A. brasiliensis* (strain CBS 101740/IMI 381727/IBT 21946)	No
**A0A9W6DJU7 (AA: 382)**	unreviewed	*A. brasiliensis*	No

Reviewed: This entry has been manually annotated by UniProtKB/Swiss-Prot, a manually annotated and non-redundant protein sequence database. Unreviewed: this is an unreviewed entry belonging to the computer-annotated TrEMBL section of UniProtKB (the section is associated with computationally generated annotation and large-scale functional characterization); AA: number of amino acid residues (only for non-full-length protein sequences).

**Table 6 antibiotics-14-00943-t006:** Crystallographic data of experimentally determined structures selected for protein-ligand docking (P-LD) simulations.

Bacterial Target: DNA Gyrase Subunit B (GyrB)					
IDs (Reference)	Organism	Resolution (Å)	Seq. Length (AA Pos)	Mutation (Mut Pos)	Co-Cry Lig (ID)
4GGL/ Q839Z1 [29]	*E. faecalis* V583	1.69	642 AA (18–224)	0	PubChem CID 70699420 (CJC) ^DI^
6TCK/ P0A0K8 [30]	*S. aureus*	1.60	644 AA (2–234)	0	PubChem CID 151595514 (N1N)
7PTF/ Q9I7C2 [31]	*P. aeruginosa* PAO1	1.32	806 AA (1–221)	0	Novobiocin (NOV)
7P2M/ P0AES6 [31]	*E. coli* K-12	1.16	804 AA (1–220)	0	PubChem CID 151595514 (N1N)
**Fungal Target: Lanosterol 14α-Demethylase Cytochrome P450 (CYP51)**					
**IDs** **(Reference)**	**Organism**	**Resolution** **(Å)**	**Seq. Length** **(AA Pos)**	**Mutation** **(AA Pos)**	**Co-Cry Lig** **(ID)**
5V5Z/ P10613 [32]	*C. albicans* SC5314	2.90	528 AA (1–528)	0	Itraconazole (1YN)
5FSA/ P10613 [33]	*C. albicans*	2.86	528 AA (48–528)	1 (221)	Posaconazole (X2N)
5TZ1/ P10613 [33]	*C. albicans*	2.00	528 AA (48–528)	2 (6, 221)	Oteseconazole (VT1)
4WMZ/ A6ZSR0 [34]	*S. cerevisiae* YJM789	2.05	530 AA (1–530)	0	Fluconazole (TPF)

IDs (Reference): UniProtKB entry/RCSB PDB identifier (reference for the RCSB PDB identifier selected for P-LD); Seq. length (AA pos): total length of the enzyme described by the UniProtKB entry (actual amino acid residues position covered by the corresponding RCSB PDB identifier—crystal structure); Mutation (mut pos): number of mutations present in the crystal structure (corresponding mutated amino acid residues—if any); Co-cry Lig (ID): PubChem Compound Identifier—CID, or the mostly known common name of co-crystallized ligand (unique RCSB PDB identifier of the ligand). ^DI^: CJC is a pyrrolopyrimidine dual inhibitor of GyrB (4GGL, *E. faecalis* V583) and topoisomerase IV (4HYM, *Francisella tularensis* subsp. *holarctica* LVS) [35].

**Table 7 antibiotics-14-00943-t007:** Docking results for bacterial GyrB. Binding affinities (BAs) were expressed in kcal/mol.

4GGL (*E. faecalis* V583)		6TCK (*S. aureus*)		7PTF (*P. aeruginosa* PAO1)		7P2M (*E. coli* K12)	
T-LC	BA	T-LC	BA	T-LC	BA	T-LC	BA
**1a**	−8.2	**1a**	−9.5	**1a**	−9.4	**1a**	−8.3
**1b**	−8.4	**1b**	−8.7	**1b**	−9.1	**1b**	−8.4
**1c**	−8.2	**1c**	−8.6	**1c**	−8.5	**1c**	−8.4
**1d**	−8.9	**1d**	−8.2	**1d**	−9.2	**1d**	−8.5
**1e**	−9.0	**1e**	−9.3	**1e**	−9.6	**1e**	−9.1
**1f**	−8.3	**1f**	−8.8	**1f**	−8.9	**1f**	−8.2
**1g**	−8.0	**1g**	−8.3	**1g**	−9.1	**1g**	−8.2
Ciprofloxacin	−8.4	Ciprofloxacin	−7.6	Ciprofloxacin	−7.7	Ciprofloxacin	−7.4
Novobiocin	−3.4	Novobiocin	−7.0	Novobiocin ^RD^	−7.6	Novobiocin	−6.0

T-LC: target—ligand complex; BA: binding affinity (expressed in kcal/mol); ^RD^: re-docking of co-crystallized ligand. RMSD (root mean square deviation): Autodock Vina calculates root mean square deviation values relative to the best binding mode and uses only the movable heavy atoms; moreover two variants of RMSD metrics are provided: RMSD lower bound (RMSD/lb) and RMSD upper bound (RMSD/ub), differing in how the atoms are matched in the distance calculation [36]

**Table 8 antibiotics-14-00943-t008:** Docking results for fungal CYP51. Binding affinities (BAs) were expressed in kcal/mol.

5V5Z (*C. albicans* SC5314)		5FSA (*C. albicans*)		5TZ1 (*C. albicans*)		4WMZ (*S. cerevisiae* YJM789)	
T-LC	BA	T-LC	BA	T-LC	BA	T-LC	BA
**1a**	−10.3	**1a**	−10.6	**1a**	−10.3	**1a**	−10.2
**1b**	−10.2	**1b**	−10.8	**1b**	−10.3	**1b**	−10.2
**1c**	−10.4	**1c**	−10.4	**1c**	−10.3	**1c**	−10.1
**1d**	−9.9	**1d**	−10.1	**1d**	−9.9	**1d**	−8.8
**1e**	−11.8	**1e**	−11.8	**1e**	−11.0	**1e**	−11.6
**1f**	−10.2	**1f**	−10.6	**1f**	−10.1	**1f**	−9.9
**1g**	−10.2	**1g**	−10.4	**1g**	−9.9	**1g**	−9.7
Fluconazole	−7.3	Fluconazole	−7.2	Fluconazole	−7.0	Fluconazole ^RD^	−7.5

T-LC: target—ligand complex; BA: binding affinity (expressed in kcal/mol); ^RD^: re-docking of co-crystallized ligand. RMSD (root mean square deviation): Autodock Vina calculates root mean square deviation values relative to the best binding mode and uses only the movable heavy atoms; moreover two variants of RMSD metrics are provided: RMSD lower bound (RMSD/lb) and RMSD upper bound (RMSD/ub), differing in how the atoms are matched in the distance calculation [36].

**Table 9 antibiotics-14-00943-t009:** The results of the molecular dynamics simulation on complexes of 7PTF.

Complex	Average Ligand RMSD (nm)	Average Backbone RMSD (nm)	Average Radius of Gyration (nm)	Average Ligand-Protein Hydrogen Bonds (No/Ns)
**1a-7PTF**	0.20	0.18	1.70	3.52
**1b-7PTF**	0.37	0.20	1.70	0.46
**1c-7PTF**	0.75	0.24	1.71	0.49
**1d-7PTF**	0.16	0.19	1.70	3.84
**1e-7PTF**	0.46	0.19	1.71	1.96
**1f-7PTF**	0.20	0.19	1.70	3.30
**1g-7PTF**	0.27	0.23	1.70	3.90
**NOV-7PTF**	0.21	0.17	1.70	4.31
**Apo 7PTF**	N/A	0.21	1.70	N/A

Legend: NOV—novobiocin.

**Table 10 antibiotics-14-00943-t010:** The computed in silico pharmacokinetic descriptors for compounds **1a**–**g**. The prediction was conducted using SwissADME web tool.

Compound	GI Absorption	BBB Permeation	P-Gp Substrate	CYP1A2 Inhibitor	CYP2C19 Inhibitor	CYP2C9 Inhibitor	CYP2D6 Inhibitor	CYP3A4 Inhibitor
**1a**	Low	No	No	No	No	Yes	No	No
**1b**	Low	No	No	No	No	Yes	No	No
**1c**	Low	No	No	No	No	Yes	No	No
**1d**	Low	No	No	No	No	Yes	No	No
**1e**	Low	No	No	No	No	Yes	No	No
**1f**	Low	No	No	No	No	No	No	No
**1g**	Low	No	No	No	No	No	No	No

**Table 11 antibiotics-14-00943-t011:** The computed in silico toxicologic descriptors for compounds **1a**–**g**. The prediction was conducted using admetSAR 2.0 web tool and Toxtree 3.1.0 software.

Compound	Carcinogenicity	Eye Irritation	Skin Irritation	Hepatotoxicity	Respiratory Toxicity	Reproductive Toxicity	Nephrotoxicity	Acute Oral Toxicity
**1a**	No	No	No	Yes	Yes	Yes	No	Class III
**1b**	No	No	No	Yes	Yes	Yes	No	Class III
**1c**	No	No	No	Yes	Yes	Yes	No	Class III
**1d**	No	No	No	Yes	Yes	Yes	No	Class III
**1e**	No	No	No	Yes	Yes	Yes	No	Class III
**1f**	No	No	No	Yes	Yes	Yes	No	Class III
**1g**	No	No	No	Yes	Yes	Yes	No	Class III

**Table 12 antibiotics-14-00943-t012:** Summary of the in vitro antioxidant activity of compounds **1a**–**g** [22].

Compound	Antiradical Assays		Electron Transfer Capacity Assays				
	IC_50_ DPPH^•^ (μM)	IC_50_ ABTS^•+^ (μM)	TAC	RP		FRAP	CUPRAC
			Eq Ascorbic Acid	Eq Ascorbic Acid	Eq Trolox	Eq Trolox	Eq Trolox
**1a**	29.90	12.62	1.53	1.96	1.52	1.37	3.09
**1b**	29.54	11.77	1.53	1.84	1.43	1.40	3.16
**1c**	29.88	11.60	1.47	1.83	1.42	1.38	3.21
**1d**	33.49	14.04	1.61	1.08	0.84	1.14	2.27
**1e**	28.60	10.88	1.33	1.71	1.33	1.20	2.99
**1f**	24.57	8.38	2.20	2.54	1.98	1.45	3.63
**1g**	23.84	7.06	2.18	2.59	2.02	1.42	3.60
**Ascorbic acid**	50.17	-					
**Trolox**	36.69	16.57					

Legend: IC_50_—half-maximal inhibitory concentration; DPPH^•^—2,2-diphenyl-1-picrylhydrazyl; ABTS^•+^—2,2′-azino-bis(3-ethylbenzothiazoline-6-sulfonic acid); TAC—Total Antioxidant Capacity; RP—Reducing Power; FRAP—Ferric Reducing Antioxidant Potential; CUPRAC—Cupric Reducing Antioxidant Capacity; Eq—equivalent molar activity.

## Data Availability

The data presented in this study are available in this article.

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
