# Peer review of "Antimicrobial and Antibiofilm Activities of Some Antioxidant 3,4-Dihydroxyphenyl-Thiazole-Coumarin Hybrid Compounds: In Silico and In Vitro Evaluation"

_antibiotics, 2025, doi:10.3390/antibiotics14090943_

Round 1
Reviewer 1 Report
Comments and Suggestions for Authors
The authors have performed antimicrobial and antioxidant activity on their previously reported compounds. In terms of antibacterial activity, most of the compounds showed moderate to promising antimicrobial activity on selective bacterial/fungal strains and found bacteriocidal and fungicidal. The in vitro results were correlated with the molecular docking studies and found with high binding affinity. Drug likeleness predictions were also performed using in silico tools. Following are the observations:
- The chemistry part is underdeveloped as it has already been published by the authors.
- The in vitro antimicrobial activity is not supported by the cytotoxicity and other assays to further confirm the potential of these compounds.
- The molecular docking is not supported by the MD simulation studies and biochemical assays.
Author Response
Responses for Reviewer #1
We would like to thank the reviewer for taking the time to assess our manuscript and for the suggestions, comments, and recommendations about the article “Antimicrobial Activity of Some Antioxidant 3,4-Dihydroxyphenyl-Thiazole-Coumarin Hybrid Compounds: In Silico and In Vitro Evaluation”. We agree with the suggestions and have made a major revision for improving the presentation of our manuscript.
Observation 1: The chemistry part is underdeveloped as it has already been published by the authors.
Response 1: We thank you for your observation. We understand that the chemistry part was underdeveloped in this manuscript, since we wanted to avoid repetitive information and possible auto-plagiarism accusations. The study of both biological activities for these compounds was initially designed to be included in a single paper. However, the extensive length of the initial draft affected its readability. Therefore, the draft was split and the chemistry part was presented in the first paper on these compounds. Potential readers interested in the chemistry of these compounds are encouraged to check out that article, since it was mentioned multiple times in the current paper.
Observation 2: The in vitro antimicrobial activity is not supported by the cytotoxicity and other assays to further confirm the potential of these compounds.
Response 2: We thank you for your observation and we understand the importance of cytotoxicity studies for the development of compounds with biological potential. In this study, we aimed to analyze the antimicrobial profile of the tested compounds and to evaluate in silico the potential mechanisms of action and pharmacokinetic and toxicological properties. We intend to complete the studies regarding the bioactive profile of the synthesized compounds with cytotoxicity studies and other assays, including the antitumoral and antibiofilm potential (work in progress).
Observation 3: The molecular docking is not supported by the MD simulation studies and biochemical assays.
Response 3: We thank you for your observation. We managed to extend the current study with a molecular dynamics simulation on one of the targets evaluated in the molecular docking. We considered that the assays performed to investigate the antioxidant and antimicrobial activities of compounds 1a-g highlighted a positive correlation between the antioxidant and antibacterial activities of these compounds. This observation represented the starting point for conducting a molecular dynamics simulation on the GyrB subunit from P. aeruginosa PAO1 (7PTF), for which the best BAs were obtained in the molecular docking study.
We updated the manuscript accordingly and the graphs resulted from the simulations are available in the Supplementary Materials.
Reviewer 2 Report
Comments and Suggestions for Authors
This article summarizes the antimicrobial and antioxidant activities of 3,4-dihydroxy-2 phenyl-thiazole-coumarin hybrid compounds against various bacterial and fungal species, as well as molecular docking studies of the same compounds against various previously identified cellular targets of antimicrobial agents. The authors also presented the effect of various functional groups attached to the coumarin moiety on the antimicrobial activity and the binding affinities. Although the work presented in the Manuscript on this timely and very important issue will attract the attention of researchers from a wide area of interest, and thus it has particular importance; there are several major issues that need to be addressed before the final publication of the work:
- The antimicrobial activity data based on the MIC values of various substituted derivatives revealed the importance of -OH substitution on the coumarin ring. Moreover, the hybrid structure scaffold contains 3,4-dihydroxy-2 phenyl moiety on the opposite arm. Although phenol group is known to be an important scaffold to target the bacterial or fungal cell membrane, authors did not consider the possibility of cell membrane targeting by these molecules. The main mode of action achieved by these molecules may very well be the disruption of the bacterial or fungal cell membrane integrity. This possibility must be evaluated by PI uptake and/or possible leakage of small molecules or ions.
- It is not clear how cellular targets of the compounds in bacterial cells and fungal cells were determined. Why did the authors focus on the coumarin binding targets for bacterial cells, whereas azole functionality was considered for antifungal activity assessment? Why not vice versa, or what about the dihydroxy phenyl group binding? More in depth discussion based on the molecular docking or other data evaluating the binding between similar structures and the cellular targets chosen must be provided.
- For a better correlation between the antimicrobial activity and binding affinity data, molecular docking of the compounds on GyrB target from a bacterial specie where low antimicrobial activities were obtained must be studied and BA values must be compared with the current values.
Author Response
Manuscript ID: antibiotics-3794765
Type of manuscript: Article
Title: Antimicrobial Activity of Some Antioxidant 3,4-Dihydroxyphenyl-Thiazole-Coumarin Hybrid Compounds: In Silico and In Vitro Evaluation
Authors: Daniel Ungureanu, Gabriel Marc, Mihaela Niculina Duma, Radu Tamaian, BrînduÈ™a Tiperciuc, Cristina Moldovan, Ioana IonuÈ›, Anca Stana, Ovidiu Oniga
Responses for Reviewer #2
We would like to thank the reviewer for taking the time to assess our manuscript and for the suggestions, comments, and recommendations about the article “Antimicrobial Activity of Some Antioxidant 3,4-Dihydroxyphenyl-Thiazole-Coumarin Hybrid Compounds: In Silico and In Vitro Evaluation”. We agree with the suggestions and have made a major revision for improving the presentation of our manuscript.
Observation 1: The antimicrobial activity data based on the MIC values of various substituted derivatives revealed the importance of -OH substitution on the coumarin ring. Moreover, the hybrid structure scaffold contains 3,4-dihydroxy-2 phenyl moiety on the opposite arm. Although phenol group is known to be an important scaffold to target the bacterial or fungal cell membrane, authors did not consider the possibility of cell membrane targeting by these molecules. The main mode of action achieved by these molecules may very well be the disruption of the bacterial or fungal cell membrane integrity. This possibility must be evaluated by PI uptake and/or possible leakage of small molecules or ions.
Response 1: We thank you for your observation and we appreciate the suggestions regarding the new potential mechanisms of action. The hypothesis of this study was based on the idea of clubbing two different pharmacophores with antimicrobial properties into a single compound. The selected pharmacophores are found in authorized compounds with well-known mechanisms: the coumarin heterocycle from novobiocin, a compound with antibacterial activity due to the inhibition of GyrB subunit, respectively the azole heterocycle from reported DNA gyrase inhibitors with antibacterial activity and from the large and important class of antifungal azoles with lanosterol 14α-demethylase inhibitory activity. Additionally, we wanted to correlate the previously reported antioxidant activity of these compounds with their antimicrobial potential.
For a better understanding of the hypothesis of this study, we updated the Introduction section with additional clarifications.
Observation 2: It is not clear how cellular targets of the compounds in bacterial cells and fungal cells were determined. Why did the authors focus on the coumarin binding targets for bacterial cells, whereas azole functionality was considered for antifungal activity assessment? Why not vice versa, or what about the dihydroxy phenyl group binding? More in depth discussion based on the molecular docking or other data evaluating the binding between similar structures and the cellular targets chosen must be provided.
Response 2: We thank you for your observation. We have stated in the previous response the hypothesis regarding our selection of the targets studied. Additionally, we included a molecular dynamics simulation study that helps providing a better insight regarding the binding of these compounds to the bacterial target.
Observation 3: For a better correlation between the antimicrobial activity and binding affinity data, molecular docking of the compounds on GyrB target from a bacterial specie where low antimicrobial activities were obtained must be studied and BA values must be compared with the current values.
Response 3: We thank you for your observation. Lower antibacterial activities were obtained against Salmonella spp. and Listeria monocytogenes, but unfortunately in the case of S. typhimurium, only the reviewed full-length sequence was identified in UniProtKB, without any corresponding experimentally determined crystal structure in RCSB PDB. For S. enteritidis and L. monocytogenes, only unreviewed full-length sequences were identified in UniProtKB, but they lack experimentally determined structures in RCSB PDB, excluding their suitability for molecular docking. This was the reason why we did not perform molecular docking studies on the aforementioned bacterial species.
Reviewer 3 Report
Comments and Suggestions for Authors
Since the aim of the study was to investigate the antimicrobial activity of previously synthesized compounds that exhibit antioxidant activity, I consider it would be appropriate in the conclusion of the abstract a) to underline only about their antimicrobial activity or b) write about correlation between antioxidant and antimicrobial activities in this part
Author Response
We thank you for your observation. We updated the conclusion of the abstract to underline only the antimicrobial activity: “This study confirmed that compounds 1a-g possess antimicrobial activity with various intensities, depending on the tested strain”.

Round 2
Reviewer 1 Report
Comments and Suggestions for Authors
As suggested, the authors have revised the manuscript partially. The study is based on a previously reported series of compounds by the same research group. The chemistry part along with some biological activities were reported earlier and the current study is based on their antimicrobial (MIC, MBC, MFC) and in silico, through molecular docking, molecular dynamics simulations, and ADMET predictions. In response to my comment, the authors have commented that:
We intend to complete the studies regarding the bioactive profile of the synthesized compounds with cytotoxicity studies and other assays, including the antitumoral and antibiofilm potential (work in progress).
The present study is incomplete and does not provide a mechanistic approach to deliver some bioactive molecules which should be thoroughly examined including the biochemical studies.
In my opinion, the authors should revise it extensively and include the data which they are doing.
Reviewer 2 Report
Comments and Suggestions for Authors
All points raised by this Reviewer are addressed.
Author Response
We would like to thank the reviewer for taking the time to assess our manuscript and for the positive review report about the article.
Round 3
Reviewer 1 Report
Comments and Suggestions for Authors
The authors have revised the manuscript and added antibiofilm activity data which has improved the manuscript significantly. Thus it can be considered for publication.